# 2,5-Hexanedione Affects Ovarian Granulosa Cells in Swine by Regulating the *CDKN1A* Gene: A Transcriptome Analysis

**DOI:** 10.3390/vetsci10030201

**Published:** 2023-03-07

**Authors:** Yige Chen, Chengcheng Kong, Min Yang, Yangguang Liu, Zheng Han, Liming Xu, Xianrui Zheng, Yueyun Ding, Zongjun Yin, Xiaodong Zhang

**Affiliations:** 1Anhui Provincial Laboratory of Local Animal Genetic Resource Conservation and Bio-Breeding, College of Animal Science and Technology, Anhui Agricultural University, No. 130, West Changjiang Road, Hefei 230036, China; 2Anhui Province Key Laboratory of Aquaculture & Stock Enhancement, Fishery Institute of Anhui Academy of Agricultural Sciences, Hefei 230031, China

**Keywords:** 2,5-Hexanedione(2,5-HD), granulosa cells, *CDKN1A*, apoptosis, sow

## Abstract

**Simple Summary:**

Porcine ovarian granulosa cells (pGCs), the main somatic cells in the follicular microenvironment, are close to oocytes. Under the action of various regulatory factors and exogenous substances, their abnormal proliferation, apoptosis and oxidative stress directly affect the quality of oocytes and embryonic development. Although 2,5-hexanedione (2,5-HD), a metabolite of n-hexane, has been linked to animal fertility in recent studies, the underlying mechanisms are still not clearly known. In this research, we aim to investigate the effects of different concentrations of 2,5-HD on cell morphology and apoptosis in pGCs. RNA-seq analyses revealed 4817 differentially expressed genes (DEGs) following 2,5-HD treatment. GO enrichment and KEGG pathway analysis suggested that the DEG, *CDKN1A*, was significantly enriched in the p53 signaling pathway that had the functions of apoptosis, growth inhibition and inhibition of cell cycle progression. After the interference of the *CDKN1A* gene, we found that it decreased pGC apoptosis, with lower cells in the G1 phase and higher cells in the S phase. Through a 2,5-HD cytotoxicity study and transcriptome bioinformatics analysis, these results deepen our understanding of the toxicological effects of 2,5-HD on porcine ovarian granulosa cells (pGCs), which is expected to help to understand the toxic mechanism of 2,5-HD in the female animal reproductive system.

**Abstract:**

N-hexane, a common industrial organic solvent, causes multiple organ damage owing to its metabolite, 2,5-hexanedione (2,5-HD). To identify and evaluate the effects of 2,5-HD on sows’ reproductive performance, we used porcine ovarian granulosa cells (pGCs) as a vehicle and carried out cell morphology and transcriptome analyses. 2,5-HD has the potential to inhibit the proliferation of pGCs and induce morphological changes and apoptosis depending on the dose. RNA-seq analyses identified 4817 differentially expressed genes (DEGs), with 2394 down-regulated and 2423 up-regulated following 2,5-HD exposure treatment. The DEG, cyclin-dependent kinase inhibitor 1A (*CDKN1A*), according to the Kyoto Encyclopedia of Genes and Genomes enrichment analysis, was significantly enriched in the p53 signaling pathway. Thus, we evaluated its function in pGC apoptosis in vitro. Then, we knocked down the *CDKN1A* gene in the pGCs to identify its effects on pGCs. Its knockdown decreased pGC apoptosis, with significantly fewer cells in the G1 phase (*p* < 0.05) and very significantly more cells in the S phase (*p* < 0.01). Herein, we revealed novel candidate genes that influence pGCs apoptosis and cell cycle and provided new insights into the role of *CDKN1A* in pGCs during apoptosis and cell cycle arrest.

## 1. Introduction

N-hexane is a type of volatile organic solvent that is mainly utilized in industrial printing, electronic device cleaning, oil extraction, and leather adhesion [1]. Inhaled n-hexane can be absorbed and can then be transported to lipid-rich tissues and organs such as the brain, peripheral nerves, liver, spleen, and kidneys [2,3,4]. Long-term exposure to n-hexane causes severe neuropathy in both humans and experimental animals [5]. An earlier study showed that the body’s primary metabolite of n-hexane was 2,5-hexanedione (2,5-HD) [6]. However, present studies on the toxicity of 2,5-HD have mainly focused on neurotoxicity [7,8,9] and testicular toxicity [10].

Despite the fact that previous studies have shown that n-hexane has harmful effects on the central and peripheral nervous systems [11,12,13], its toxic mechanisms on the female reproductive system have remained unclear. The ovary is an important reproductive organ of mammals, and an abnormal ovulation process will directly affect the reproductive performance of female animals. The proliferation, apoptosis, and functional differentiation of ovarian granulosa cells are of great significance to the ovary, the growth and development of follicles, and the formation of the corpus luteum. Ovarian granulosa cells are essential for oocyte development [14]. N-hexane may directly mediate granulosa cell apoptosis by changing hormone secretion, which may be one of the important mechanisms of n-hexane-induced ovarian dysfunction in mice [15]. Sun et al. discovered that in human ovarian granulosa cells, *BAX* and active *CASPASE-3* (p17) expression significantly increased in a dose-dependent manner, while *BCL-2* expression reduced with increasing 2,5-HD concentrations [16]. Apoptosis was the mechanism of germ cell loss in 2,5-HD-induced testicular injury [17]. When the testes were treated with 2,5-HD, it increased the retention time of the sperm head in the semen and decreased sperm motility [18]. Exploring the mechanism for the cytotoxic effect of 2,5-HD on ovarian granulosa cells would thus prove meaningful.

In this study, we explored the toxicological effects of 2,5-HD on porcine ovarian granulosa cells (pGCs) via a 2,5-HD cytotoxicity study and RNA-seq. The findings of the present study are expected to contribute to a better understanding of the molecular biological mechanism by which 2,5-HD causes apoptosis in ovarian granulosa cells.

## 2. Materials and Methods

The entire procedure was carried out exactly in accordance with the guidelines approved by the Anhui Agricultural University Animal Ethics Committee under Permission No. AHAU20180615.

### 2.1. Porcine Ovarian Granulosa Cells (pGCs) Culture In Vitro and under 2,5-HD Treatment

Fresh porcine ovaries (*n* = 60, from Landrace, approximately 180 days old) with a bright surface, healthy development, full follicles, and more antral follicles were obtained from a commercial slaughterhouse (Hefei, Anhui, China), around which the fatty tissue was removed. The ovaries were maintained in sterile saline solution at 37 °C containing 1% penicillin/streptomycin mixture and delivered to the lab within 1 h in a thermos flask. Avoiding blood vessels, the follicular fluid was collected from ovarian follicles (*n* = 300, 3–6 mm in diameter) [19] by using the injector (1 mL), washed, centrifuged (1000× *g*, 5 min), resuspended, and centrifuged again, and the pGCs were then harvested. The cells were cultivated at 37 °C and 5% CO_2_ in Dulbecco’s modified Eagle medium/Nutrient Mixture F-12 medium (DMEM/F12, Gibco, Grand Island, NY, USA) containing 10% fetal bovine serum (FBS) (Gibco, Carlsbad, CA, USA), 100 units/mL penicillin, and 100 mg/mL streptomycin (Gibco, Carlsbad, CA, USA) after being seeded into a 60 mm cell-culture dish (Corning, 430166, Somerville, MA, USA). 

2,5-HD was purchased from Macklin (Shanghai, China), and its purity was determined to be greater than 99.5%. The 2,5-HD solution was dissolved in pure water to create a 100 mol/mL stock solution. The primary pGCs could be passaged at approximately 80% confluence (the primary pGCs completely stretched and occupied approximately 80% of the cell culture dish surface; about one week), and the pGCs were inoculated in a 12-well plate (Corning, 430166, USA) at a density of 8~10 × 10^5^ cells as calculated by the cell count plates (Watson, 177–122c, Kobe, Japan) and allowed to attach for 12 h. Flow cytometry was used to detect the purity of pGCs (the cells’ purity degree was > 90%). Then, the cells were treated with 2,5-HD (0 mmol/L, 20 mmol/L, 40 mmol/L, 60 mmol/L) at 37 °C for 24 h, and the same growth cell culture medium was used to treat the control group (three biological repeats in both the treatment and control groups). The dose and reagent treatment time used in this experiment were based on previous studies that revealed the toxic effects on rat and sows’ ovarian granulosa cells [20,21] in vitro using this concentration and time.

### 2.2. Cell Morphological Observations

After treatment with 2,5-HD (0 mmol/L, 20 mmol/L, 40 mmol/L, 60 mmol/L) for 24 h, we used a microscope (Leica Microsystems DM2500, Wetzlar, Hessen, Germany) to examine the cellular morphology photographs.

### 2.3. Cell Apoptosis Detection

The pGCs were treated with 2,5-HD (0 mmol/L, 20 mmol/L, 40 mmol/L, 60 mmol/L) for 24 h and were collected according to the instructions of the cell flow kit (Annexin V-FITC/PI,BB-4101, Bestbio, Shanghai, China) [22] and pretreated. The pGCs were collected, pretreated, trypsinized for 1 min, centrifuged (1000× *g*, 3 min), washed twice with PBS, resuspended in 500 µL of binding buffer containing 5 µL Annexin V-fluorescein isothiocyanate (FITC) and 10 µL propidium iodide (PI) (Bestbio, Shanghai, China). Finally, a FACS Calibur flow cytometry device (FACSCalibur, BD, Franklin Lakes, NJ, USA) was used to calculate the quantity of stained cells.

### 2.4. Phalloidin Staining

The pGCs were inoculated in a 12-well plate and allowed to attach for 12 h. Cells were then treated with 2,5-HD media (0 mmol/L, 20 mmol/L, 40 mmol/L, 60 mmol/L) at 37 °C for 24 h. After about 45 min of incubation at room temperature in a dark area, an antifade substance containing fluorescein isothiocyanate (FITC) was injected into the cells, which were then rinsed with PBS for 10 min. FITC-phalloidin (Solarbio CA1620, Beijing, China) was added for 30 min of re-staining, followed by 10 μL of anti-fluorescence quenching tablets containing 4′,6-diamidino 2-phenylindole (DAPI) to obtain FITC-phalloidin-labeled cytoskeleton and DAPI-labeled cellular nuclei. Finally, a fluorescent microscope (Leica Microsystems DM2500, GER) was used to view the cells.

### 2.5. EDU Staining

EDU (5-ethynyl-2′-deoxyuridine) staining enables quick and effective cell proliferation assays to accurately measure the proportion of cells in the S phase. Therefore, using EDU staining in accordance with the manufacturer’s instructions, the proliferation of pGCs was determined. After treatment with 0 mmol/L, 20 mmol/L, 40 mmol/L, and 60 mmol/L 2,5-HD for 24 h, cells were cultured in phenol red-free DMEM/F12 with 50 μM EdU (RiboBio, Guangzhou, China) staining for 2 h [23,24]. Cells were then rinsed two times with 0.02 M PBS (pH = 7.2) at 5-min intervals before being immobilized for 30 min with 4% paraformaldehyde (Biosharp, Hefei, China). Following that, cells were treated for 5 min with glycine (Biosharp, Hefei, China) (2 mg/mL) and rinsed for 5 min with PBS [23,24]. After 30 min of DAPI (0.5 μg/mL) (Biosharp, Hefei, China) staining [23], three rounds of PBS washing were conducted at 5-min intervals. Lastly, a Leica DM4000 BLED microscope (Leica Microsystems DM2500, GER) was used to observe the fluorescence in cells.

### 2.6. Total RNA Extraction, Library Construction, and Sequencing Analysis

With the above results, we proceeded to extract RNA and performed transcriptome sequencing. pGCs were plated into a 55-cm^2^ cell culture dish and grown in DMEM/F12 with or without 2,5-HD (40 mmol/L) for 24 h. As stated in our previous report, the treatment and control groups’ total RNA was extracted [25]. Thereafter, the RNA was purified with the RNeasy Mini Kit (Qiagen, Hilden, Germany) and delivered to Majorbio (Shanghai, China). The Nanodrop2000 and Agilent 2100 systems were employed to determine total RNA concentration, purity, and integrity. RNA qualities were found to meet the requirements for database sequencing (Table 1). The mRNA was then fragmented with a fragmentation buffer after being enriched with oligo dT. The cleaved RNA fragments were then reverse-transcribed to create the final cDNA library. We connected an Illumina Hiseq. 4000 [23] (Illumina, San Diego, CA, USA) to an adaptor and carried out paired-end sequencing in accordance with the recommended procedure from the manufacturer. Using the DESeq R program (1.18.0), differentially expressed genes (DEGs) were found. Both values of a fold change (log2FC) and the FDR (FDR = *p*-value corrected for multiple hypothesis tests) were used to filter the significant or insignificant DEGs. These statistical procedures use a model based on the negative binomial distribution to identify differential expression among digital gene expression data. To reduce the rate of false discovery, Benjamini and Hochberg’s method was used to adjust the resulting *p*-values, and an adjusted *p* < 0.05 was deemed statistically significant.

Gene ontology (GO; http://www.geneontology.org/, accessed on 28 October 2022) and the Kyoto Encyclopedia of Genes and Genomes (KEGG; http://www.genome.jp/kegg, accessed on 5 November 2022) were used to find the enriched pathways and the statistically enriched genes. 

### 2.7. Quantitative PCR

Quantitative real-time PCR was used as proof that high-throughput sequencing was accurate. To detect the expression trend of the sequencing results, 12 genes were chosen at random. TRIzol reagent (Invitrogen Corporation, Carlsbad, CA, USA) was used to extract the cells’ total RNA, and the Primer-Script RT reagent Kit (TaKara, Tokyo, Japan) was used to reverse-transcribe the extracted total RNA into cDNA. The CFX96 Touch Real-Time PCR Detection System (Bio-Rad, Hercules, CA, USA) was used for qRT-PCR analysis. Based on sequence data of the NCBI database (www.ncbi.nlm.nih.gov/genbank, accessed on 20 November 2022), the specific primers utilized for RT-PCR were designed or modified by Primer Premier 6.0 software (Premier Biosoft International, Palo Alto, CA, USA) [26] and are listed in Table 2. The thermal cycling program consisted of 95 °C for 30 s, followed by 40 cycles at 95 °C for 5 s and 60 °C for 30 s. The housekeeping gene glyceraldehyde-3-phophate dehydrogenase (*GAPDH*) was used to standardize the expression levels of the genes, and the changes in relative gene expression were computed by the 2^−ΔΔCt^ method. Significant differences were evaluated by Student’s *t*-test using SAS software (version 9.0), and a value of *p* < 0.05 was considered to be significant, which was calculated at least three times from independent biological replicates.

### 2.8. CDKN1A Interference

For interference of *CDKN1A*, small interfering RNA (siRNA) was purchased from Ribobio (Guangzhou, China) (Table 3). pGCs in the logarithmic phase were collected and inoculated at a density of 8~10 × 10^5^ on a 6-well plate. According to the instructions of the transfection kit (Ribobio, Guangzhou, China) [27,28], siRNA-*CDKN1A*, negative control (siRNA-NC), and 12 μL riboFECT CP Reagent [29,30] (Ribobio, Guangzhou, China) were co-incubated with cells. Then, 48 h after transfection, the pGCs were collected. Thereafter, cells were used for RNA extraction, measuring the cell apoptotic rate and cell cycle detection. Transfection was carried out utilizing the Lipofectamine 3000 reagent (Invitrogen, Waltham, MA, USA).

### 2.9. Statistical Analysis

All experiments were performed in triplicate, and all data were expressed as mean ± standard deviation (SD)with Student’s *t*-test. One-way ANOVA was carried out to determine significant differences in the experiment data in triplicate. The statistical analyses were performed with SAS software Version 8.01 (SAS Institute Inc.; Cary, NC, USA). Statistical significance was defined as *p* < 0.05, with very significant findings classified as *p* < 0.01 and extremely significant findings as *p* < 0.001. 

## 3. Results

### 3.1. Effect of 2,5-HD on pGC Morphology

We randomly selected five regions to obtain the cell morphology results. As shown in Figure 1, the granulosa cells of the control group were fusiform or irregularly polygonal, fully stretched, tightly arranged, translucent, and displayed adequate adhesion. The cells displayed dendritic connections with an increase in the mass concentration of 2,5-HD (24 h, 40 mmol/L). When the concentration of 2,5-HD was increased to a certain degree (24 h, 60 mmol/L), the pGCs grew spherically, and some cells were detached and suspended.

After that, actin skeletons were stained with FITC-phalloidin (green), and cell nuclei were stained with DAPI (blue). The control group granulosa cells were attached and spread, accompanied by discrete bundles of actin fibers. Less prominent expansions to neighboring cells that remained clustered and an unorganized cytoskeleton were present in the 2,5-HD-treated group of cells (Figure 2).

### 3.2. Effect of 2,5-HD on Apoptosis of pGCs

The results of Annexin V-FITC/PI double staining and flow cytometry showed that the effect of 2,5-HD treatment on the apoptotic rate of pGCs was dose-dependent, and the 20 mmol/L (24 h) treatment group had a significantly higher rate of pGC apoptosis than that the control (*p* < 0.05). The difference in the apoptotic rate became even more pronounced when the 2,5-HD concentration was at 40 mmol/L (24 h) and 60 mmol/L (24 h) (*p* < 0.01) (Figure 3).

### 3.3. Effect of 2,5-HD on pGC Proliferation

Cellular and chromatin ultrastructures that have been maintained well can be stained with EDU to identify cell proliferation. We randomly selected five regions to obtain the staining results. Images were first captured, and the results obtained with the images were used to perform a difference analysis. As shown in Figure 4, the percentage of EDU-positive cells was significantly lower in the 2,5-HD-treated group (20 mmol/L, 24 h) than in the control group (*p* < 0.01). Proliferation-free cells were discovered in the randomly selected fields of the 40 mmol/L (24 h) group and the 60 mmol/L (24 h) group. These results demonstrated that a 40 mmol/L (24 h) 2,5-HD treatment inhibited the proliferation of pGCs. After pGCs were exposed to 2,5-HD, chromatin condensation and a larger nucleus were discovered, as shown in Figure 5 by DAPI staining.

Analysis of cell morphology, proliferation, and apoptosis revealed that a treatment period of 24 h at a dose of 40 mmol/L (2,5-HD) was ideal for further study. In fact, after 24 h of treatment, the pGCs also displayed dendritic connections with an increase in the mass concentration of 2,5-HD, the apoptotic rate changes reached very significant levels between the 2,5-HD-treated group (40 mmol/L, 24 h) and the normal control group (*p* < 0.01), and no proliferating cells were found in the 2,5-HD-treated group (40 mmol/L, 24 h). Finally, we chose 0 mmol/L and 40 mmol/L for the RNA-seq analysis experiments that followed (three biological replications). 

### 3.4. Gene Expression Profiling

The sequencing results showed 8,262,103,500 and 11,571,998,400 bp raw reads in each group. Q20 (base sequencing error probability < 1%) > 97% and Q30 (base sequencing error probability < 0.1%) > 94% were achieved for each group, indicating good data quality (Table 4). Six sample gene expression patterns were used for principal component analysis (PCA), and sample correlations were projected onto the first two principal components (Figure 6). Three biological replicates of gene expression showed a consistent pattern, and the sequencing quality turned out to meet the standards for additional analyses.

### 3.5. Analysis of DEGs

According to the DEGs discovered in the 2,5-HD (0 mmol/L and 40 mmol/L for 24 h) treatment group of pGCs, gene and pathway analyses were conducted. A total of 22,955 genes were obtained from the RNA-seq results by comparing them to the reference genome from Scrofa11.1 of Ensembl (http://www.ensembl.org/, accessed on 16 October 2022). The significant DEGs’ relative transcription levels were shown by a fold change |log_2_FC| > 1 and FDR < 0.05 (Figure 7A); a fold change |log_2_FC| ≤ 1 and FDR ≥ 0.05 was regarded as insignificant. Nonetheless, the analysis showed up to 4817 DEGs (Appendix A). Of these, 2423 were up-regulated, while 2394 were down-regulated (Figure 7B).To verify the reliability of sequencing data, six genes (*IRAK2*, *SLC7A11*, *SPP1*, *CDKN1A*, *CASP3* and *MDM2*) that were significantly up-regulated in the treatment group (2,5-HD 40 mmol/L, 24 h) and six genes (*MYCL*, *RENBP*, *DKK2*, *F3*, *MYH11* and *BCLAF1*) that were significantly down-regulated in the normal control group (2,5-HD 0 mmol/L, 24 h) were randomly selected for RT-qPCR verification. These qRT-PCR results aligned with the sequencing results, indicating the reliability of RNA-Seq data (Table 5 and Figure 8). 

### 3.6. KEGG and GO Analysis

For a more in-depth view of the identified DEGs, we conducted GO and KEGG pathway enrichment analyses. The three GO categories—biological process, cellular component, and molecular function—were used to group together all of the GO terms associated with DEGs. The results of the enriched analysis revealed that, for the biological process, cellular component, and molecular function categories, 67, 26, and 13 GO terms were significantly enriched (*p* < 0.05) (Appendix A), respectively. The DEGs were mainly involved in different developmental processes, including intracellular, organelle, and binding. The top 20 enriched GO terms are shown in Figure 9A, which demonstrates that the majority of the biological process are linked to cell activity, such as the cell cycle (Appendix A). In KEGG enrichment, we found the p53 signal pathway was one of the important metabolic pathways with the smallest Q value (Q value= *p*-value following corrections from multiple hypothesis tests) (Figure 9B and Appendix A). The activation of the p53 signaling pathway is mainly induced by DNA damage, oxidative stress, and oncogene expression and plays a positive role in promoting cell apoptosis, regulating the cell cycle, and preventing tumorigenesis [31]. Moreover, the PI3K/Akt signaling pathway, with its ovarian-relevant functions of the recruitment of primordial follicles, the proliferation of granulosa cells (GCs), the formation of the corpus luteum, and oocyte maturation [32], was also significantly enriched (Q value < 0.05) (Figure 9B and Appendix A). 

It should be noted that, through function analysis, we discovered that the *CDKN1A* gene was significantly up-regulated in 2,5-HD-exposed pGCs compared with the control group, which was involved in apoptosis, the regulation of the cell cycle, and follicular development. Then, we used interference of *CDKN1A* for transfection to reduce its expression to further demonstrate the regulation of *CDKN1A* gene on pGCs.

### 3.7. Effect of CDKN1A on Apoptosis of pGCs

RNA interference was utilized to reduce *CDKN1A* expression in pGCs cultivated in vitro to investigate the role of *CDKN1A* in the apoptosis of pGCs. By qPCR analysis, *CDKN1A* expression was significantly lower in pGCs transfected with the *CDKN1A*-siRNA than it was in pGCs transfected with the NC-siRNA (*p* < 0.05) (Figure 10A). After transfection, the rate of apoptosis was lower in the *CDKN1A*-siRNA group than in the *CDKN1A*-NC group according to flow cytometry analysis and a quantitative histogram; nevertheless, the difference was not statistically significant (*p* > 0.05) (Figure 10B,C).

### 3.8. Effect of CDKN1A on the Cycle of pGCs

To investigate the periodic distribution of cells in the transfected and control groups, pGCs were measured by flow cytometry for 48 h after transfection. As shown in Figure 11, the G1 phase had a significantly lower number of cells (*p* < 0.05), while the S phase had a very significantly higher number of cells compared to the control group (*p* < 0.01). These findings revealed that *CDKN1A*-siRNA could promote the transition of pGCs from the G1 phase to the S phase, decrease cell cycle arrest of pGCs in the G1 phase, and increase pGC proliferation.

## 4. Discussion

pGCs are an important class of ovarian cells located outside the zona pellucida. Granulocytes play a vital role in the regulation of ovarian function, and prior research has demonstrated that granulosa cells perform a critical function in the regulation of follicular atresia [12,33]. As the apoptosis of granulosa cells may have a major impact on reproductive efficiency [34], the apoptosis of pGCs may play an important role in the study of follicular atresia and oocyte development. The toxicity of 2,5-HD has been extensively studied [21]. In fact, previous studies have revealed that it destroys female reproductive function and leads to a decrease in fertility [35,36]. 2,5-HD can significantly prolong the estrous cycle of female rats, with the ovary serving as one of the potential target gonads of n-hexane toxicity [37].

Through the present study, we demonstrated that treatment of pGCs with 2,5-HD can cause morphological changes. In particular, we revealed that it could increase the apoptotic rate of cells and inhibit cell proliferation. A previous study revealed an increase in the apoptosis of human neuroblastoma SK-N-SH cells exposed to 2,5-HD (3.3 +/− 10.1 mM) [38]. Additionally, other studies have demonstrated that HD (500 nM similar to 50 mM) dose-dependently suppresses the proliferation and viability of murine neural progenitor cells (NPCs) and increases the production of reactive oxygen species (ROS). HD (10 or 50 mg/kg for 2 weeks) inhibited hippocampal neuronal and NPC proliferation in male 6-week-old ICR mice, primary neuronal culture, and young adult mice [12]. The study findings indicated that 2,5-HD (5 and 10 mmol/L) could decrease the viability of pheochromocytoma cells (PC12) and promote apoptosis via oxidative injury in a concentration-dependent manner [39]. Mitochondrial-dependent apoptosis was also found to be induced by HD through NGF suppression, which occurs via the PI3K/Akt pathway, both in vivo and in vitro [40]. The results of the present investigation supported earlier findings about morphological alterations and the impacts of 2,5-HD on apoptosis and proliferation.

According to the 2,5-HD cytotoxicity experiments with pGCs and RNA-seq, we obtained important information, including the ten genes with the highest differential expression (the 10 with the highest fold change value). Sulfiredoxin (*SRXN1*) was previously found to be associated with cerebrovascular disease in a Finnish cohort [41], and *SRXN1* genetic polymorphisms were associated with breast cancer risk and survival [42]. *SRXN1* is, therefore, necessary for resolving GnRH-induced oxidative stress and inducing gonadotropin gene expression [42]. Wang et al. [43] found that *CCNG2* encoded an unconventional cyclin homolog, cyclin G2(*CycG2*), which is associated with growth inhibition and significantly correlated with lymph node metastasis, histological grade, and poor overall survival in numerous cancer types. The *ANKRD1* mutations may cause DCM as a result of disruption of the normal cardiac stretch-based signaling [44]. The *GREM1* gene is associated with diabetic nephropathy [45], while Id3 governs the downstream mitogenic processing via depressing p21(*WAF1/Cip1*), p27(*Kip1*), and p53 [46]. A previous study showed that *CCL2* is an inflammatory mediator with proinflammatory activity in breast cancer [47]. These known functions support our conclusion of the effects of 2,5-HD.

The GO enrichment results revealed that the DEGs were primarily connected to developmental processes, cellular developmental processes, cell proliferation, cell cycle, cellular component organization, and other important terms. Furthermore, 11 significantly enriched pathways were also found, including the p53 signaling pathway, MAPK signaling pathway, renin secretion, amino sugar and nucleotide sugar metabolism, PI3K-Akt signaling pathway, and other important metabolic pathways. The p53 signaling pathway, activated by DNA damage, oxidative stress, and proto-oncogene activity, plays important roles in promoting apoptosis, regulating the cell cycle, and preventing tumorigenesis [31,48,49]. N-hexane can induce or enhance the production of oxygen free radicals in organisms, damage lipid peroxidation, and induce DNA damage in rat hepatic cells [50]. Studies have shown that oxidative stress damage from exposure to n-hexane and its metabolite 2,5-HD can damage ovarian cells, which will affect the endocrine function of the ovary [51]. An earlier study found that 2,5-HD promoted apoptosis in mesenchymal stem cells and proposed that the activated mitochondria-dependent caspase-3 pathway may be involved in 2,5-HD-induced apoptosis [1]. The p53-induced Siva-1 gene expresses an effector molecule that plays a significant role in DNA damage-induced cell death [31]. High doses of 2,5-HD induce apoptosis by activating the p53 pathway through genetic toxicity and oxidative stress effects. The major up-regulated genes enriched in the p53 signaling pathway are activators of the apoptosis execution factors; the *PMAIP1* gene directly binds to *Bax* or *Bcl2* antagonist/killer 1 to increase human granulosa cell proapoptotic activity [52]. The overexpression of *IGFBP3* can promote cell proliferation and migration [53]. The *CASP3* gene can cause morphological and biochemical changes in the cells of the body, resulting in apoptosis [54,55]. The phosphatidylinositol 3-kinase (PI3K)-Akt signaling pathway can be activated by various types of cellular or toxic stimuli, thereby regulating basic cellular functions, including transcription, translation, proliferation, growth, and survival [56,57]. A prior study also showed that 2,5-HD down-regulates rat spinal nerve growth factor and induces neuronal apoptosis by inhibiting the PI3K-Akt signaling pathway [58]. In this study, the cell cycle-driving *cyclin* proteins (*CCND1*, *CCND3*) were significantly down-regulated and enriched within the PI3K-Akt pathway, which inhibited follicular development by inhibiting cell cycle progression and granulosa cell proliferation.

The p53 signaling pathway was found to be significantly enriched in this study, and previous research indicated that it is the primary pathway involved in apoptosis. Interestingly, *CDKN1A*, also known as p21 (*WAF1/CIP1*), was demonstrated to be involved in the cell cycle [59]; such a finding attracted our attention in the current study. *CDKN1A* is implicated in the regulation of cell growth and cell response to DNA damage [60]. The p21 protein may function during development as an inducible growth inhibitor that contributes to cell cycle exit and differentiation [61]. circGRAMD1B plays an important role in GC progression by regulating miR-130a-3p-PTEN/p21 [62]. NeuroD1-silencing induces the expression of p21, a master regulator of the cell cycle, leading to G2-M phase arrest and the suppression of colorectal cancer cell proliferation and the potential of colony formation [63]. *CDKN1A* was also selected as an important candidate gene affecting cell cycle and apoptosis. Therefore, exogenous stimulation caused pGC DNA damage, and p21(*CDKN-1A*) was involved in apoptosis in a p53-dependent approach.

*CDKN1A* modulates the cell cycle, apoptosis, senescence, and differentiation via specific protein–protein interactions with cyclins and cyclin-dependent kinase (Cdk) [64]. In the present study, *CDKN1A* was demonstrated to induce the apoptosis of pGCs and cell cycle arrest in G1. However, the apoptosis rate of pGCs after transfection was not significantly different compared to the control group (*p* > 0.05). According to the previous reports, it was found that cell cycle uncoupling occurred after the knockout of the p21 gene in the human colon adenocarcinoma cell line HCT116, which was treated with multiple DNA damage agents [65]. Therefore, we speculated that the insignificant difference between *CDKN1A*-siRNA and *CDKN1A*-NC in the apoptotic rate of pGCs may be due to the uncoupling of the cell cycle caused by the inhibition of p21 (*CDKN1A*) expression, which leads to polyploid giant cells returning to the S phase and, eventually, death. These results were consistent with our cell cycle experiment: *CDKN1A* interference promoted the percentage of cells in the S phase. Meanwhile, the relationship between exogenous p21 expression and apoptosis was controversial. P21 induced apoptosis in some cells, while resisting p53-induced apoptosis in others [66]; the specific mechanisms and principle need to be further studied. Moreover, a chemically induced fibrosarcoma model was utilized to demonstrate that p53 and *CDKN1A* cooperate in mediating cancer resistance [67]. Strong p21 accumulation suppresses the CDK inhibitory activity of E2F-dependent transcriptional processes and cell cycle arrest in G1 [68,69]. 

## 5. Conclusions

In conclusion, we demonstrated that 2,5-HD could affect the proliferation and apoptosis of pGCs in pigs. Furthermore, by performing transcriptome analysis, we found that the *CDKN1A* gene was significantly up-regulated in 2,5-HD-induced pGCs, and its knockdown could increase the number of cells in the S phases of the cell cycle. Altogether, our findings indicate that the *CDKN1A* gene may play important roles in the apoptosis of 2,5-HD-induced pGCs.

## Figures and Tables

**Figure 1 vetsci-10-00201-f001:**
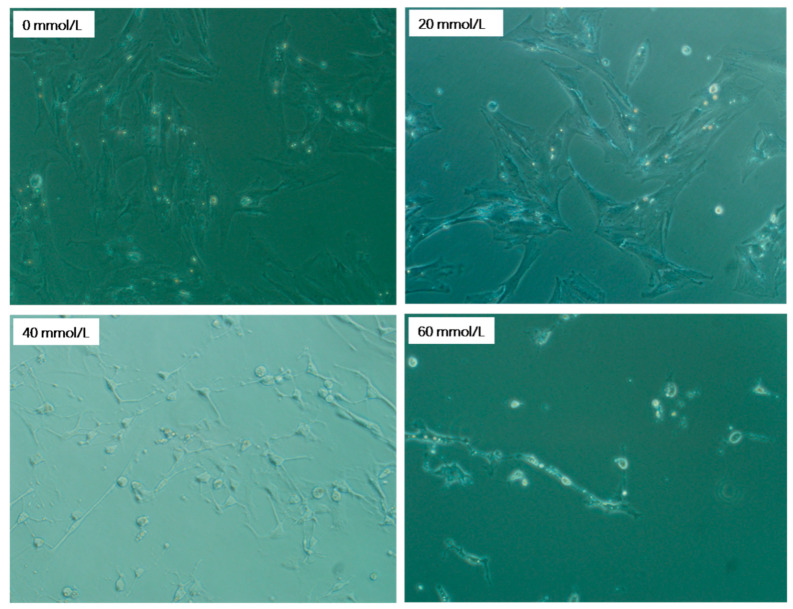
Cell morphology observed by fluorescence microscopy. The granulosa cells of the porcine ovary (pGCs) were exposed to various concentrations of 2,5-HD (0 mmol/L, 20 mmol/L, 40 mmol/L and 60 mmol/L) for 24 h. Scale bar 200 μm.

**Figure 2 vetsci-10-00201-f002:**
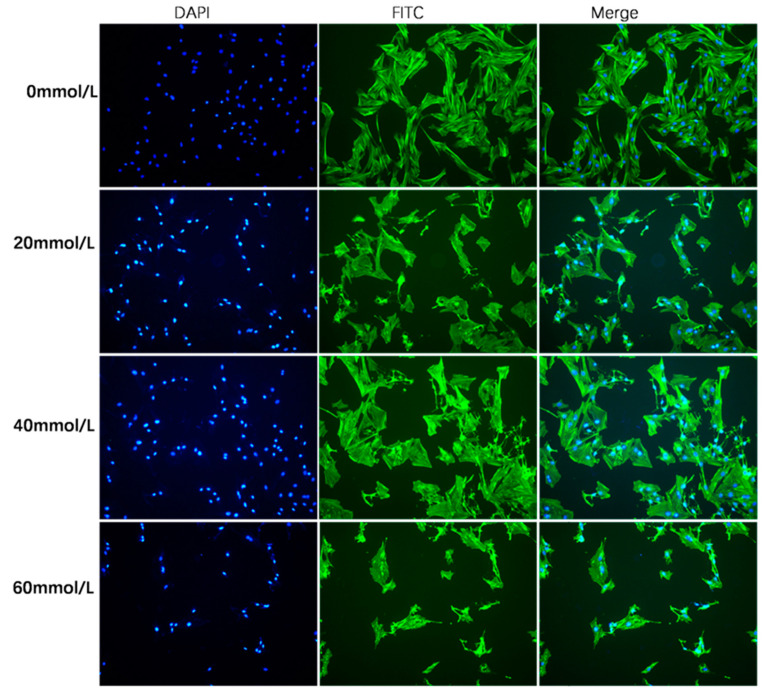
Cytoskeleton morphology image from fluorescence microscopy. Laser scanning confocal microscopy was used to observe the granulosa cells of the porcine ovary. Using the fluorescence immunocytochemistry method, cells were stained with various doses of 2,5-HD (0 mmol/L, 20 mmol/L, 40 mmol/L and 60 mmol/L) for 24 h. Scale bar 200 μm.

**Figure 3 vetsci-10-00201-f003:**
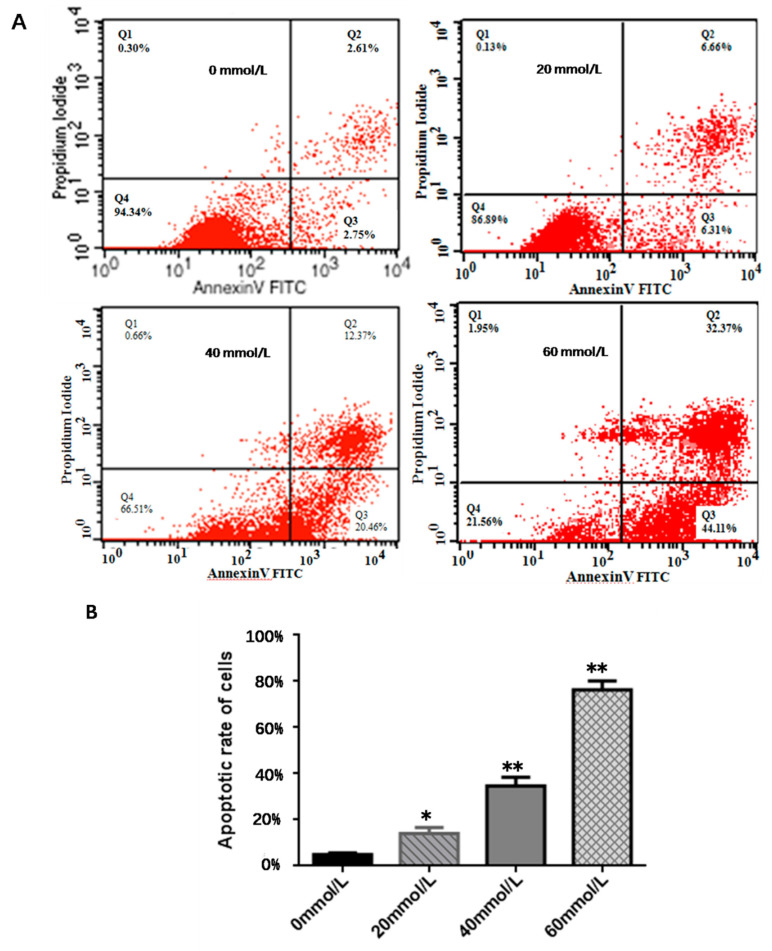
pGC apoptosis detection. (**A**) Effects of different concentrations of 2,5-HD (0 mmol/L, 20 mmol/L, 40 mmol/L, and 60 mmol/L) for 24 h on pGC apoptosis by flow cytometry. (**B**) Effects of different concentrations of 2,5-HD (0 mmol/L, 20 mmol/L, 40 mmol/L and 60 mmol/L) for 24 h on pGC apoptosis. Compared to the control group, *n* = 3, * *p* < 0.05, ** *p* < 0.01. All experiments were repeated 3 times.

**Figure 4 vetsci-10-00201-f004:**
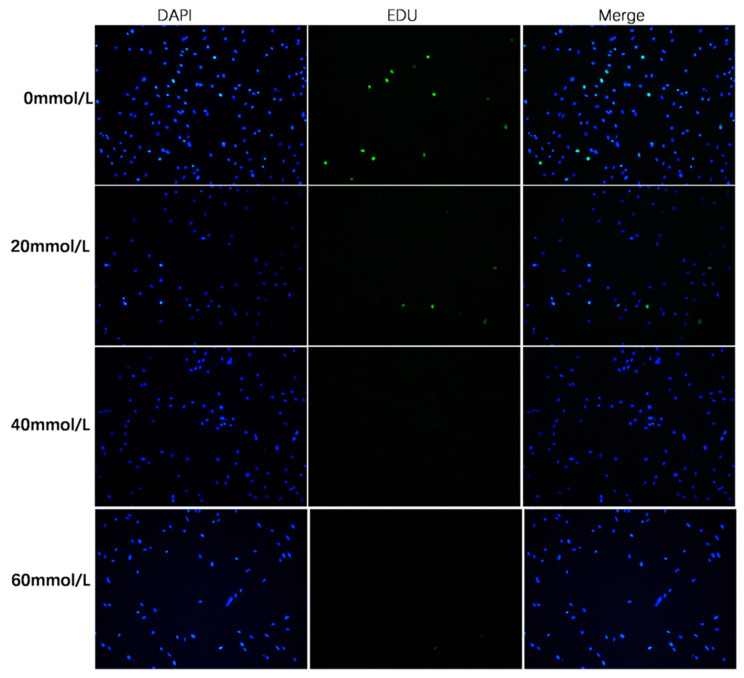
The detection of pGC proliferation by fluorescence microscopy. Following a 24-h treatment with different doses of 2,5-HD (0 mmol/L, 20 mmol/L, 40 mmol/L and 60 mmol/L), the proliferation of pGCs was assessed using an EDU assay. DAPI was used to color the cell nuclei blue, and EDU-positive cells were colored green. Scale bar 200 μm.

**Figure 5 vetsci-10-00201-f005:**
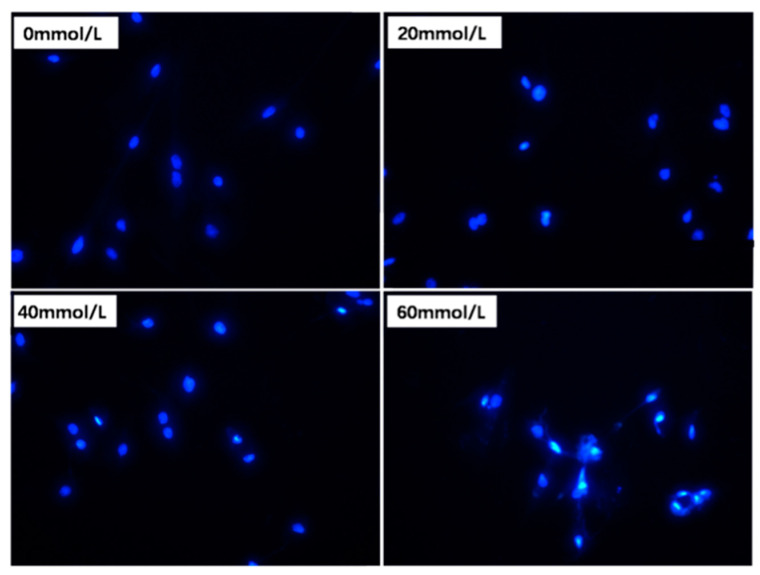
2,5-HD-induced nuclei changes in pGCs detected by fluorescence microscopy. The pGCs’ nuclear division following treatment with 2,5-HD (0 mmol/L, 20 mmol/L, 40 mmol/L and 60 mmol/L) for 24 h was detected by DAPI staining. Scale bar 200 μm.

**Figure 6 vetsci-10-00201-f006:**
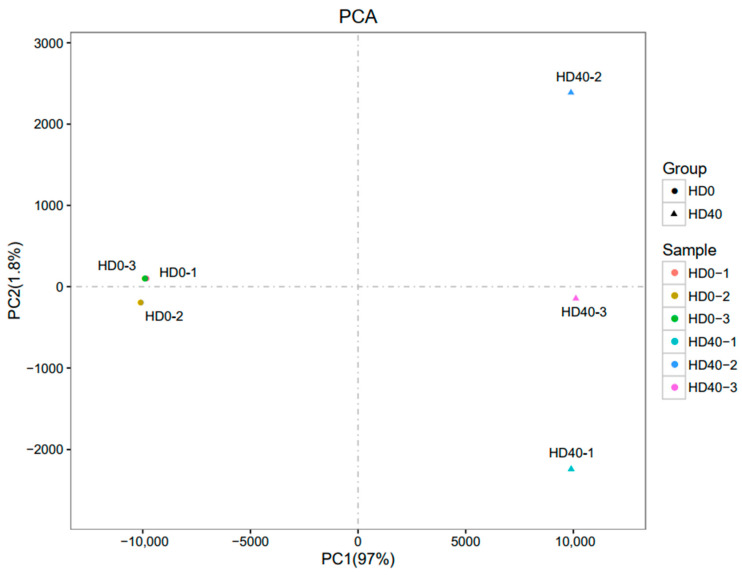
Principal component analysis (PCA) of samples’ relationship. The control groups (HD0-1,2,3) are represented by the triangle nodes, while the 2,5-HD treated groups (HD40-1,2,3) are represented by the circle nodes. The two principal components, PC1 and PC2, together accounted for 98.8% of the total variance, with PC1 accounting for 97% and PC2 for 1.8%.

**Figure 7 vetsci-10-00201-f007:**
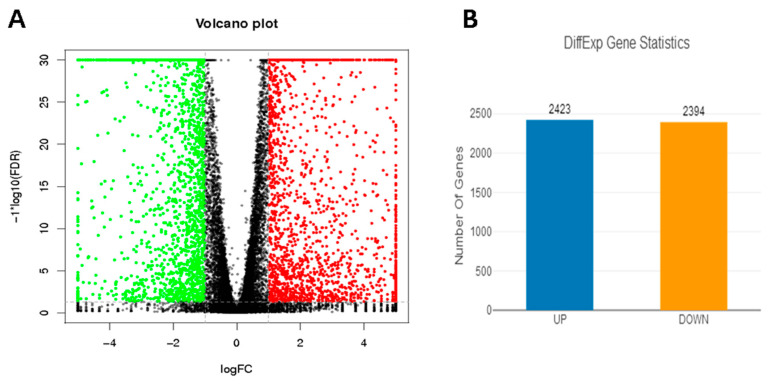
The identification of differentially expressed genes (DEGs). (**A**) DEGs with higher (H) and lower (L) levels are displayed on the volcano plot. Genes that differ significantly are represented by red and green points, respectively, with a fold change |log_2_FC| >1 and FDR < 0.05. Genes with a fold change |log_2_FC| ≤ 1 and no significant difference are represented by black points; red and green points denote up-regulated and down-regulated genes, respectively. (**B**) Higher (H) and lower (L) levels of DEG statistics. Down-regulated and up-regulated genes are represented on the *X*-axis, while the number of DEGs is represented on the *Y*-axis.

**Figure 8 vetsci-10-00201-f008:**
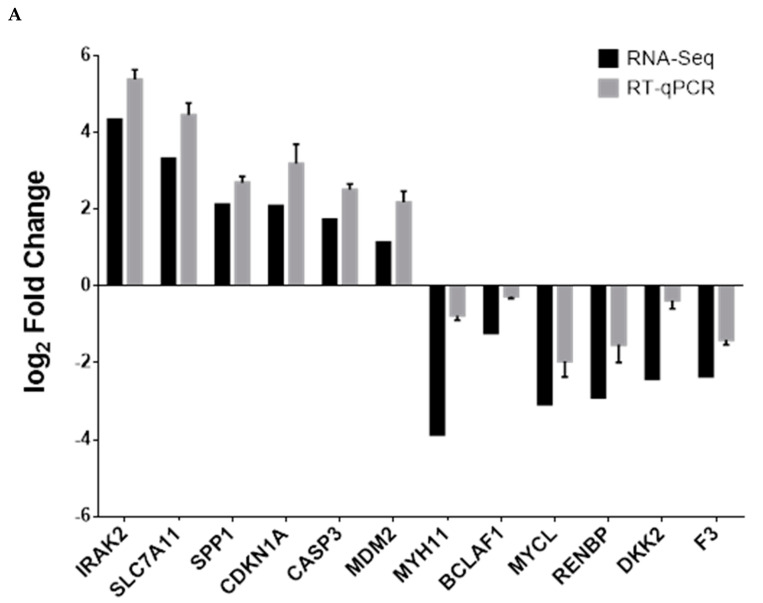
(**A**) qPCR validation of DEGs. A comparison of RNA-Seq and qPCR validation results. The *X*-axis represents genes in the validated comparison of this study; the *Y*-axis represents the Log2 ratio for the control and 2,5-HD (40 mmol/L, 24 h) treatment groups. (**B**) Line fit plot of qRT-PCR results and RNA-Seq data showing the linear regression model, R-Squared and expression difference of the selected 12 DEGs (the blue points) between the treatment group (40 mmol/L, 24 h) and the control group (0 mmol/L, 24 h). The *X*-axis and the *Y*-axis represent the log_2_ ratio for RNA-Seq and RT-qPCR, respectively, between which correlation trend was represented by the blue dotted line All experiments were repeated 3 times.

**Figure 9 vetsci-10-00201-f009:**
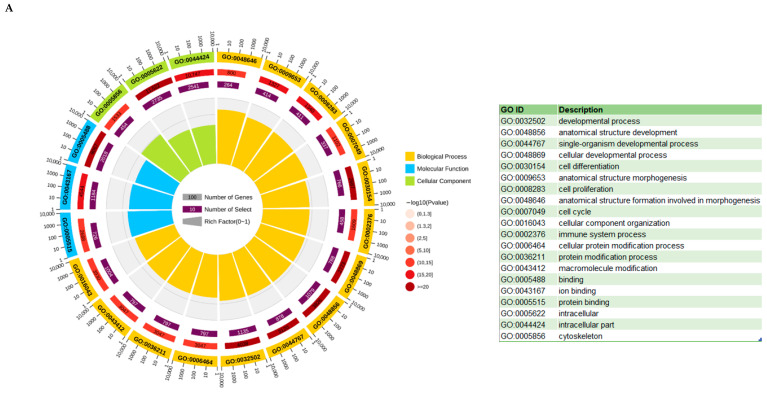
Function enrichment analysis of transcriptomic data. (**A**) Top 20 enriched GO terms of the DEGs. The first lap indicates the top 20 GO terms, and the number of the genes corresponds to the outer lap. The second lap indicates the number of genes in the genome background and *p*-value for enrichment of the differentially expressed genes (DEGs) for the specified biological process. The third lap indicates the DEG number. The fourth lap indicates the rich factor of each GO term. (**B**) Significantly enriched KEGG pathways from differentially expressed genes (DEGs). The first lap represents the significantly enriched pathways’ ID, and the number of genes corresponds to the outer lap. The second lap represents the number of genes in the genome background as well as the Q-value for pathways. The third lap represents the DEG number. The fourth lap shows the rich factor of each KEGG pathway. *P*-value *<* 0.05 and Q value < 0.05 were considered significant in GO and KEGG analyses, respectively.

**Figure 10 vetsci-10-00201-f010:**
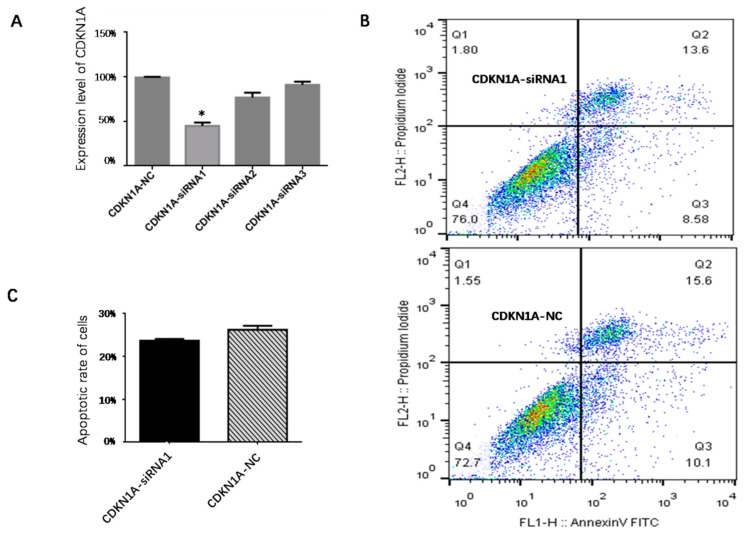
(**A**) Quantification map of transfection efficiency, * *p* < 0.05. (**B**) Effects of *CDKN1A*-siRNA1 and *CDKN1A*-NC on pGC apoptosis by flow cytometry after transfection. (**C**) Quantitative histogram of the apoptotic rate after transfection of *CDKN1A*-siRNA1 and *CDKN1A*-NC on pGC apoptosis. All experiments were repeated 3 times.

**Figure 11 vetsci-10-00201-f011:**
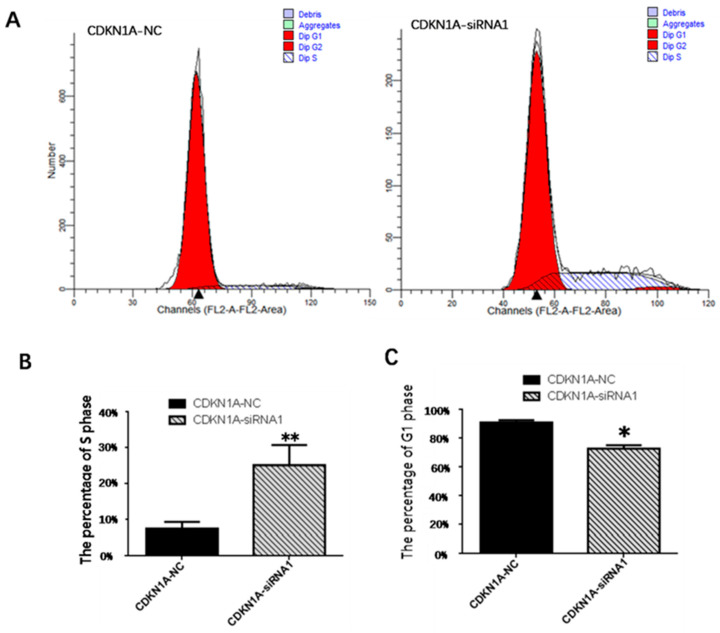
(**A**) Cell cycle analysis of pGCs after transfection of *CDKN1A*-siRNA1 and *CDKN1A*-NC. (**B**) Quantification of the S phase by flow cytometry after transfection. (**C**) Quantification of the G1 phase by flow cytometry after transfection, * *p* < 0.05, ** *p* < 0.01. All experiments were repeated 3 times.

**Table 1 vetsci-10-00201-t001:** The results of RNA extraction test.

Production Number	Sample	Concentration-ng/uL	Volume/uL	Total/μg	RIN Value	Library Type	Conclusion
R18081118	HD0-1	498	41	20.42	9.3	RNA-seq	A
R18081119	HD0-2	465	41	19.07	10.0	RNA-seq	A
R18081120	HD0-3	412	42	17.30	9.4	RNA-seq	A
R18081121	HD40-1	308	40	12.32	9.5	RNA-seq	A
R18081122	HD40-2	282	38	10.72	9.2	RNA-seq	A
R18081123	HD40-3	225	40	9.00	9.7	RNA-seq	A

Note: HD0-1~3 represents the control groups; HD40-1~3 represents the 40 mmol/L treatment groups, and the groups are added for 24 h; the three repetitions are denoted by the numbers 1, 2 and 3; the letter A represents that the quality and total amount meet the requirements of subsequent experiments.

**Table 2 vetsci-10-00201-t002:** Primer sequences used in qPCR experiments.

Gene Symbol	Primer Sequence (5′-3′)	Up/Down
*MYH11*	GCGTCCATGCCAGATAACAC	Down
	CGCCCGACTTTGATACGTG	
*MYCL*	GGACCCCTGCATGAAACACT	Down
	CTGCCTCCTCTTCCTTTTCG	
*RENBP*	GAGTGGGCCATGAAGCTCT	Down
	CGAAACTGGCGAAACGTGTA	
*DKK2*	TTTGCTGTGCACGTCACTTC	Down
	TTCTTGCGCTGCTTGGTACA	
*F3*	ACGCCCTACCTGGACACAAA	Down
	TGCCGTTCACCCTGACTAAG	
*BCLAF1*	GATTCGGAAGGGGATGACAC	Down
	CCTCCTCAGTATTCCGGTGA	
*MDM2*	ACGACAAAGAAAGCGCCACA	Up
	ACTCACACCAGCGTCGAGAT	
*CASP3*	CGGACAGTGGGACTGAAGAT	Up
	CGCCAGGAATAGTAACCAGG	
*CDKN1A*	GACCATGTGGACCTGTTGCT	Up
	GGCGTTTGGAGTGGTAGAAA	
*SPP1*	AGAGACCCTGCCAAGCAAGT	Up
	ATGAGACTCGTCGGATCGGT	
*SLC7A11*	TATCTCTGGCATTTGGACGC	Up
	GCACTCCAGCTGACACTCA	
*IRAK2*	GCTCAGGTCCAGGATTGATTG	Up
	GCCCAGCAGAGGTAAGATGTT	
*GAPDH*	ATTCCACCCACGGCAAGTT	*GAPDH*-F
	TTTGATGTTGGCGGGATCT	*GAPDH*-R

**Table 3 vetsci-10-00201-t003:** Target sequence of the siRNA.

Gene	Target Sequence
si-ssc-CDKN1A_001	CCAGCATGACAGATTTCTA
si-ssc-CDKN1A_002	CCAAACGCCGGCTGATCTT
si-ssc-CDKN1A_003	GCCGGCTGATCTTCTCCAA

Note: *CDKN1A*-siRNA sequence is the target sequence; NC-siRNA was synthesized by Guangzhou Reebok Science and Technology Biology Co., Ltd., Guangzhou, China, and its sequence is confidential.

**Table 4 vetsci-10-00201-t004:** Results of RNA-seq data quality analysis.

Sample	Before Filter					After Filter				
	Clean	Q20 (%)	Q30 (%)	N (%)	GC (%)	HQ Clean	Q20 (%)	Q30 (%)	N (%)	GC (%)
	Data (bp)					Data (bp)				
HD0-1	8,546,605,500	8,344,655,111 (97.64%)	8,047,846,082 (94.16%)	1,288,743 (0.02%)	4,550,166,659 (53.24%)	8,224,210,048	8,107,811,882 (98.58%)	7,867,744,147 (95.67%)	857,487 (0.01%)	4,374,980,230 (53.20%)
HD0-2	10,701,057,900	10,470,640,137 (97.85%)	10,123,251,944 (94.60%)	1,615,274 (0.02%)	5,683,390,145 (53.11%)	10,322,343,616	10,187,549,091 (98.69%)	9,903,589,358 (95.94%)	1,073,648 (0.01%)	5,478,669,969 (53.08%)
HD0-3	9,492,620,100	9,284,411,677 (97.81%)	8,972,299,225 (94.52%)	1,432,089 (0.02%)	5,053,549,739 (53.24%)	9,147,813,875	9,026,488,425 (98.67%)	8,771,971,839 (95.89%)	949,030 (0.01%)	4,866,908,860 (53.20%)
HD40-1	8,278,856,700	809,754,7348 (97.81%)	7,826,959,983 (94.54%)	1,251,797 (0.02%)	4,324,792,272 (52.24%)	7,978,267,014	7,872,909,409 (98.68%)	7,652,604,412 (95.92%)	830,497 (0.01%)	4,163,722,726 (52.19%)
HD40-2	11,571,998,400	11,316,248,473 (97.79%)	10,935,522,921 (94.50%)	1,738,051 (0.02%)	6,103,038,102 (52.74%)	11,151,334,819	11,003,287,936 (98.67%)	10,693,484,565 (95.89%)	1,160,264 (0.01%)	5,875,747,248 (52.69%)
HD40-3	8,262,103,500	8,082,046,730 (97.82%)	7,811,943,104 (94.55%)	1,245,894 (0.02%)	4,320,317,791 (52.29%)	7,967,279,552	7,862,124,495 (98.68%)	7,641,947,568 (95.92%)	830,065 (0.01%)	4,162,633,359 (52.25%)

Note: HD0-1~3 represents the control groups; HD40-1~3 represents the 40 mmol/L treatment groups, and the groups are added for 24 h; the three repetitions are represented by the numbers 1, 2, and 3.

**Table 5 vetsci-10-00201-t005:** Detailed information on the top 10 genes with the highest differential expression.

Gene ID	Gene Name	Control	2,5-HD	Log_2_ Fold Change	Up/Down
ENSSSCG00000007554	ZFAND2A	411.333	8812.667	4.457916894	Up
ENSSSCG00000023298	SRXN1	1613.667	24,429.000	4.006772443	Up
ENSSSCG00000008988	CCNG2	411.667	4900.333	3.610374422	Up
ENSSSCG00000001488	GCLC	1608.333	17,005.667	3.483567018	Up
ENSSSCG00000022649	SLC7A11	602.000	5672.333	3.33201828	Up
ENSSSCG00000010461	ANKRD1	15,720.333	299.333	4.606115928	Down
ENSSSCG00000033657	GREM1	6506.000	358.000	4.085838065	Down
ENSSSCG00000039514	ID3	11,693.000	722.333	3.919259823	Down
ENSSSCG00000017723	CCL2	16,245.333	1019.000	3.912958487	Down
ENSSSCG00000000146	MYH11	24,112.870	1581.530	3.875688189	Down

## Data Availability

All read data are available in the NCBI SRA database (project ID: PRJNA553614).

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
