# Peer review of "2,5-Hexanedione Affects Ovarian Granulosa Cells in Swine by Regulating the CDKN1A Gene: A Transcriptome Analysis"

_vetsci, 2023, doi:10.3390/vetsci10030201_

Round 1

Reviewer 1 Report

This study evaluates the effects of 2,5-HD on porcine ovarian granulosa cells, as a model to assess the effects of this compound on reproductive function. Methodological approaches include morphological, transcriptomic, and functional analyses to assess the toxic effects of 2,5-HD on porcine granulosa cells. Overall, the study is relevant, contributes to the advanced knowledge in the field, and was conducted rigorously. The addressed methodologies were also appropriated and performed with adequate quality controls. Nevertheless, the following minor/major considerations should be addressed:

Minor considerations:

(1) The article has several formatting issues, including no spaces between the full stop and the beginning of the next sentence and low quality figures. These issues must be corrected.

(2) During the GC isolation procedure, was the level of contamination with theca cells assessed? 

(3) Was the isolated granulosa cells' cell viability assessed prior to culture?

(4) Line 92: Indicate the Kit's reference.

(5) Line 93: Provide the centrifugation conditions in line 93.

(6) Give the reference for the reagents and the washing conditions used in lines 109 to 119. This observation applies to all reagents in which the reference is missing (for example: the transfection kit and materials in lines 163-170).

(7) Line 120–140: Describe the criteria (fold change; Log2FC...) used to filter the DEGs in the materials and methods section.

(8) Specify the number of replicates for each experiment (for instance, in the in vitro culture of ovarian granulosa cells (pGCs)).

(9) Describe the methodology used for primers design.

(10) Indicate the scale bar in the images (Figure 1; Figure 2; Figure 4 and Figure 5) and specify the acquisition method (confocal microscopy, fluorescence microscopy…) in the legend.

(11) Improve the quality of Figures 6 and 9.

(12) Line 187: Avoid non-quantifiable observations ("grew well") in line 187.

(13) Line 316: remove “red”

(14) Line 322: “expression” instead of “expre-ssion”

Major considerations:

(1) Overall, the transcriptome analysis was underlooked. The results of the GO analysis were not properly evaluated. Among the top 10 most relevant BP are GO terms that may be important to discuss in this study, namely “cell cycle” (Table S2). For these GO terms, it is important to provide information on whether they are upregulated or downregulated and the most relevant DEGs involved (relevant to this study).

(2) Although only the P53 signaling pathway was discussed in the KEGG enrichment analysis, other statistically significant pathways should also be discussed, such as the PI3K pathway, associated with follicular development. Likewise, the DEGs assigned to the most relevant pathways should be referred to, indicating whether they are upregulated or downregulated. From the way the data are presented and discussed, it appears that only the P53 signaling pathway and its CNKN1A gene are responsible for his 2,5-HD effects.

(3) Regarding the discussion, the transcriptome analysis should be addressed globally and taking into account the impact of the 2,5 HD on the granulosa cells' function. The PI3K/Akt pathway, for instance, was discussed in a context unrelated to reproductive function. In addition, not only the top 10 genes, but also the downregulation of proliferation-related DEGs and the upregulation of apoptosis-related DEGs should be discussed. 

Reviewer 2 Report

The manuscript by Yige Chen et al. entitled “2,5-Hexanedione affects ovarian granulosa cells in swine by 2 regulating the CDKN1A gene: A transcriptome analysis” is a very meaningful study. The report of Yige Chen et al in this manuscript is scientifically sound. n-Hexane, a common industrial organic solvent, causes multiple organ damage, owing to its metabolite, 2,5-hexanedione (2,5-HD). To identify and evaluate the effects of 2,5 HD on sows' reproductive performance, author used porcine ovarian granulosa cells (pGCs) as a model and carried out cell morphology and transcriptome analyses. 2,5-HD has the potential to inhibit the proliferation of pGCs and induce morphological changes, apoptosis, with depending on the dose. RNA-seq analyses revealed 4,817 differentially expressed genes (DEGs), of which 2,394 were downregulated and 2,423 were upregulated following treatment. The DEG, Cyclin-dependent kinase inhibitor 1A (CDKN1A), according to the Kyoto Encyclopedia of Genes and Genomes enrichment analysis, was significantly enriched in the p53 signaling pathway. Thus, author evaluated its function in pGCs apoptosis in vitro. Then, author knocked down the CDKN1A gene in the pGCs to identify its effects on pGCs. Its knockdown decreased pGC apoptosis, with significantly lower cells in the G1 phase (p<0.05) and very significantly higher cells in the S phase (p < 0.01). Herein, study revealed novel candidate genes that influence pGCs apoptosis and cell cycle, provided new insights into the role of CDKN1A in pGCs during apoptosis and cell cycle arrest.

This is a meaningful topic, The authors obtained efficient data from different groups of pGCs with different treatments. The experimental design and procedure are clear. The research topic falls within the scope of veterinary sciences, so publication is recommended after modification.

There were some questions as follows:

1. There is no basis for the dose of 2,5-hexanedione, just citing references is unconvincing.

2. The author's basic writing level needs to be improved, please strengthen your writing skills

For example:

CDKN1A is spelled in full when it appears for the first time on paper and abbreviated when it appears again.

Spaces should be added between words. Please check the whole paper for corrections.

3. In the figure, there are some issues that need to be noticed.

For examples:

Fig 2, the distance between images should be consistent.

Fig 5, figure notes should be aligned uniformly.

Fig 6, the control and 2,5-HD treated groups figure notes and descriptions of captions are inconsistent and should be carefully revised by the authors.

4. Why primary pGCs haven't been identified? Please explain.

5. I recommend the authors either utilize an online English editing service or collaborate with a colleague whose primary language is English to correct the grammatical errors in their manuscripts.

6. How many days old sow follicles were collected to extract GCs in this study?

7. Why not detect other apoptosis-related indicators to increase the structural integrity of the paper?

8. Fig 5, does the nucleus divide to demonstrate apoptosis?

Reviewer 3 Report

In this manuscript, Chen et al examined the effect of 2,5 HD on porcine granulosa cells using transcriptome analysis. From the transcriptome analysis, they found CDKN1A as a major player and its function was evaluated using a knockdown approach. The study is interesting however certain foundational data is missing from the paper that calls for major revision.

The very function of ovarian granulosa cells is the production of hormones. These cells are central for female reproduction. On the basis of these grounds,

Was steroidogenesis and hence hormone production affected by exposure to 2,5 HD?

What type of granulosa cells were collected? 

The authors themselves in the introduction mention that ". n-Hexane may directly mediate granulosa cell apoptosis by changing hormone secretion, which may be one of the important mechanisms of 49 n-hexane-induced ovarian dysfunction in mice" . This can very well be the mechanism here as well. This needs to be substantiated and clarified.

Reviewer 4 Report

This article is generally interesting and well written. However, before the publication some points need to be clarified.

We have no information about the sows from which the varies came. Race, age, region from which they came were not given. This seems important due to pre-slaughter exposure to toxins.

We do not have information about the size of the research group and the criteria for selecting the material. Were the ovaries randomly selected, or were there any selection criteria. The same applies to the ovarian follicles.

Table 4 is illegible. You should work on the style of presenting the results in it.

Round 2

Reviewer 1 Report

The authors made a great effort to improve the manuscript in several aspects. As a result, this version of the manuscript is considerably improved. However, there are still some issues that need to be carefully revised. The English still needs to be revised and improved, especially in the discussion section, and some of the initial formatting problems persist (the space between the full stop and the beginning of the next sentence is often missed, and there are missing spaces between words).

Minor considerations:

1- The GC isolation procedure described in the cover letter should be integrated into the material and methods section. This should be a very brief description (including the degree of purity).

2- The software used for primer design, described in the cover letter, should be integrated into the material and methods section.

3- The images should have a scale bar.

4- Line 312: Remove “were collected”

5- Line 305-346:

Overall, the previous representations of the GO and KEGG analysis were fine. The requested improvement was related to the description/interpretation of the results. Although supported by supplemental material, these new representations, despite containing valuable information, are more difficult to interpret isolated as the name associated with the GO terms and KEGG pathways are missing. I suggest that the name associated with the GO term and KEGG pathways be included in these representations.

6- Line 315: Include also the reference to Table S2 (Table S2; Table S3)

7- Line 379: The full stop is missing in “in fertility[34,35]”.

8- Lines 374, 388, 392, 399, 409, 424, 437, 447, 450,459, 461: the space after the full stop is missing. Please, revise all these formatting issues.

9- Line 417-442: Please, improve English.

10- Line 447-455. The statement that CDKN1A is also known as P21 should only be placed the first time CDKN1A is mentioned (Line 445).

11- Remove the figures/tables citation in the discussion.

Reviewer 4 Report

I believe that the reader of the manuscript deserves an explanation of where the research material comes from. Please also include information on the number of ovaries and ovarian follicles used in the experiment and each research method.
